# Varicella Seroprevalence in Healthcare Workers at a Medical Center Following Changes in National and Local Hospital Vaccination Policies

**DOI:** 10.3390/ijerph16193544

**Published:** 2019-09-22

**Authors:** Meng-Ting Tsou, Hsin-Hui Shao

**Affiliations:** The Department of Family Medicine, Mackay Memorial Hospital, Taipei City 10449, Taiwan; chocosophi@gmail.com

**Keywords:** healthcare worker, seroprevalence, varicella, immunization

## Abstract

Background: Varicella seroprevalence in healthcare workers at a tertiary care hospital in Taiwan was assessed following the inclusion of varicella zoster vaccination in the national vaccination schedule in 2004 and was made a hospital policy in 2008. Methods: Seroprevalence data were extracted from records of pre-employment health check-ups performed between 2008 and 2018 at a single medical center. Staff with complete medical records and anti-varicella zoster virus immunoglobulin G (VZV IgG) titers were included. Sex and age group differences in terms of geometric mean titer (GMT) were compared using analysis of variance and chi-squared tests. The significance of the correlation between age and the anti-VZV IgG titer was tested by linear regression. The odds of significant associations among age, sex, vocation, and the years of national and hospital adoption of vaccination were determined using univariate and multivariate analyses. *p* < 0.05 was considered statistically significant. Results: Of the 7314 eligible participants, 5625 (76.90%) were women, and the mean patient age was 26.80 ± 8.00 years. The lowest VZV-positivity rates were in 18–20-year-old women (85.16%; GMT, 362.89 mIU/mL) and men (87.59%; GMT, 288.07 mIU/mL). VZV positivity increased with age (*p* < 0.001). Participants born before 2002 were more likely to be seropositive than those born after 2003 (odds ratio, 2.51 vs. 1.0; *p* < 0.001). The lowest seropositive rate was found in the nursing staff (88.91%; 95% confidence interval, 87.74%–90.05%). Varicella vaccine boosters have been required at pre-employment health check-ups since 2008 if anti-VZV antibodies were not detectable. A follow-up evaluation found marginal significant differences in the odds ratios of seropositivity after 2007 (*p* = 0.052), especially in 2008 and 2014 (*p* < 0.05) after the hospital policy launched. Conclusions: Despite public health efforts, a small number of healthcare workers were inadequately protected, and antibody titers were lower than required to maintain herd immunity. For effective prevention of nosocomial infection, VZV IgG status should be documented for all HCWs, and susceptible HCWs should be vaccinated to avoid outbreaks. Pre-employment screening and vaccination have increased immunity and need to be conducted to ensure protection of vulnerable patients.

## 1. Introduction

It is important that the immunity of the hospital staff to varicella zoster virus (VZV) be monitored in order to decrease the risk of nosocomial infection and infection to the staff and visitors [1,2]. Varicella is highly contagious and spreads via respiratory aerosols or vesicle fluid. It is usually self-limiting but may cause severe complications [2,3,4]. Approximately 11,000 cases of varicella are reported per year in Taiwan [5]. Free VZV vaccination of children aged >1 year was adopted in 2004 [5]. After national policy for vaccination, its incidence in infants and young children has fallen sharply, but the outbreaks of varicella infection may occur in adults as the anti-VZV antibody titer falls. The estimated varicella-associated hospitalization rate is 60 per 1000 patients and is highest among those aged 19–38 or >75 years [5,6]. Therefore, a national recommendation of vaccination for susceptible healthcare workers (HCWs) has been made but not yet implemented owing to the costs of testing and vaccine administration. Several outbreaks of nosocomial varicella infection have occurred in recent years [7]. Because preventing and managing nosocomial varicella is costly, vaccination of HCWs without immunity may be the most cost-effective strategy [8,9,10,11,12].

Our hospital began routine pre-employment screening of all staff in 2008, with varicella vaccine boosters required for anyone with undetectable antibody titers. Self-reported histories of varicella infection (including native or occupational exposure) are not accepted to avoid vaccination; only serology data are considered. This study assessed the varicella seroprevalence among HCWs after the implementation of national and hospital HCW vaccination policies in Taiwan to help guide the development of a local screening program.

## 2. Methods

### 2.1. Study Center and Population

The study was performed at Mackay Memorial Hospital, a 2000-bed tertiary care hospital in Northern Taiwan, a region with an estimated population of 2.67 million. The study data were retrieved from routine pre-employment health check-ups of HCWs conducted between January 2008 and June 2018 that included screening of VZV antibodies. Inclusion criteria: (1) Age was greater than or equal to 18 years old (the legal age of majority); (2) Full-time HCWs. There were no special exclusion criteria because pre-employment health check-ups are the routine examination by government law, and HCWs needed to finish the vaccination and get the certification before they joined the workplace. There was no sampling in this study. All HCWs were explained the main goal of this research before obtaining an informed consent.
The sample size was estimated including six age groups (18–20, 21–30, 31–40, 41–50, 51–60, and 61–70 years of age) and assuming a 5% error in the seroprevalence data and a confidence interval (CI) of 95%. The age ranges were established in relation to a target cohort of VZV vaccine recipients included in the universal mass vaccination of all newborns in 2003. Older subjects were enrolled to compare the seroepidemiology of varicella in the pre- and post-vaccination eras. HCW categories included doctors (physician, surgeon, specialist, dentist, traditional Chinese physician), nurses (emergency department, outpatient clinic), examination department (laboratory, ultrasound, endoscopy, catherization), preventive and long-term care services (physical examination center, community medicine division, nursing home, daycare center), and administration.

### 2.2. Laboratory Values

A 5 mL serum sample was collected from each participant for determination of anti-varicella zoster virus immunoglobulin G (anti-VZV IgG) with a quantitative varicella IgG enzyme-linked immunosorbent assay of known sensitivity (98.42%, 95% CI = 96.25%–99.31%) and specificity (93.94%, 95% CI = 79.83%–99.34%). A concentration ≥165 mIU/mL was considered reactive (positive). Tests with borderline results (135–165 mIU/mL) were repeated.

### 2.3. Statistical Analysis

Participant age, sex, and laboratory results were entered into a FileMaker Pro (www.filemaker.com) database and analyzed by SPSS (IBM Corp., Armonk, NY, USA). Age and sex differences in the geometrical mean anti-VZV IgG titers (GMTs) were compared with Student’s *t*-test or analysis of variance. Chi-squared tests were used to compare differences in the proportions of anti-VZV IgG positive participants. The significance of correlations of age and anti-VZV IgG titer was determined by linear regression analysis. Univariate and multiple logistic regression analysis was used to determine the odds of immunity following VZV exposure associated with age, sex, vocation, the year of government and hospital vaccination policy. *p*-values of <0.05 were considered significant.

### 2.4. Ethical Certification

The study was certificated by the Ethics Committee of Mackay Memorial Hospital (No. 18MMHIS103).

## 3. Results

VZV seroprevalence was assessed in 7314 participants between 2008 and 2018 during the annual pre-employment health screening (power = 0.8, α = 0.05, β = 0.2). The mean age was 26.80 ± 8.00 (18–68) years, 5625 (76.90%) participants were women, and the men were older than the women (28.14 ± 6.98 vs. 26.40 ± 8.23; *p* < 0.001). The characteristics of the men and women in each age group are shown in Table 1. The anti-VZV IgG titer was ≥165 mIU/mL in 6629 participants (90.63%, 95% CI: 89.72–91.87) of the participants. The percentage was higher in men (*n* = 1549/1689; 91.71%; 95% CI: 90.72–92.66) than in women (*n* = 5080/5625; 90.31%, 95% CI: 89.25–91.57; *p* < 0.001). In the age groups (Figure 1), 757/885 participants (85.54%, 95% CI: 83.22–87.86) from 1820, 4330/4778 participants (90.62%, 95% CI: 89.79–91.45) from 21–30, 1011/1092 participants (92.58%, 95% CI: 91.03–94.14) from 31–40, 380/400 participants (95.00%, 95% CI: 92.86–97.14) from 41–50, 122/130 participants, (93.85%, 95% CI: 89.72–97.98) from 51–60, and all 29 (100%) of those from 61–70 years of age were seropositive (*p* < 0.001). The mean anti-VZV IgG titer was 542.90 mIU/mL and was higher in men (563.14 mIU/mL) that in women (475.49 mIU/mL, *p* < 0.001). The sex difference persisted when the analysis included the age groups (women F = 33.23, men F = 18.31; *p* < 0.001). There was significant correlation between age and anti-VZV IgG titer in both women (*r*^2^ = 0.006) and men *r*^2^ = 0.01, both *p* < 0.001, Figure 1).

The seroprevalence estimates and their 95% CIs for the various healthcare providers (vocations). are shown in Table 2. Multiple logistic regression of immunity status (Table 3) found no significant difference in the odds of immunity in male and female employees. Young employees were significantly less likely to be immune, and those who were more than 50 years of age had higher odds of immunity than elderly employees (*p* < 0.001). The adjusted odds of immunity were significantly lower for employees born in recent decades. Nursing and administrators were significantly less likely to be seropositive than employees in other vocations (OR = 0.58–0.69, *p* < 0.001). Since 2008, varicella zoster seropositivity has been assayed in hospital employees, and boosters are required for those with undetectable antibody titers. The odds ratio calculation found marginal significant differences in immunity after/before 2007 (*p* = 0.052), especially in 2008 and in 2014 (*p* < 0.05) after the booster vaccination policy was implemented, (Table 4).

## 4. Discussion

This was the first study to document the effects of national and hospital vaccination policies on VZV seroprevalence in HCWs in Taiwan. VZV seropositivity was 90.63%, which is consistent with the 91.1% previously reported in Taiwan [7]. It has been reported that a group immunity of 94% or more is needed to interrupt viral transmission in the healthcare setting [2,13]. The study results showed a lack of VZV immunity in young participants 18–20 years of age, in whom nearly 15% were not seropositive. The percentage was less than 10% in other age groups. VZV seropositivity rate in the HCWs in this hospital may not be sufficiently high to prevent outbreaks of varicella; interventions to prevent varicella transmission are desirable.

Free VZV vaccination of children >1 year of age has been included in the national vaccination schedule in Taiwan since 2004 [5]. Varicella is now a preventable disease in Taiwan, and its incidence in infants and young children has fallen sharply. The hospital records show that varicella infection decreased by 82% in 2004 and 72% in 2012 [6]. The durability of the protection provided by vaccine-induced antibodies is limited, and outbreaks of varicella infection may occur in adults as the anti-VZV antibody titer falls. The hospitalization rate of adults with varicella is approximately 44.1 per 1000 people, which is much higher than that of young varicella patients (10.4 per 1000 people) [6]. Logistic regression analysis of the 2018 data found that staff born before 2002 had significantly higher odds of being immune than those born after 2003. Our crude annual seroprevalence estimates were in keeping with the reported national rates for adults, which ranged from 88.3% at 11–20 and 99.6% at >65 years of age [14]. The available epidemiological data indicate that 90%–95% of the general population was infected by VZV before adulthood and that about 5%–10% of adults are susceptible to infection [15]. GMTs generally increased with age but fell between 560–700 IU/mL in 2–5-year-olds. The titer in those older than 13 years of age was >1000 IU/mL, indicating that the antibody titer induced by natural infection was significantly higher than that induced by vaccination [6]. As the incidence of disease continues to decrease, the chance of exposure to VZV will decrease. In the absence of a natural booster, the protection provided by early childhood vaccination may decrease in adolescence and adulthood. That may result in an increase of varicella complications in addition to the risk HCW infection [2,6].

The overall hospital-wide laboratory-confirmed seroprevalence of 90.63% (95% CI 89.81–91.89) in this study is comparable to the 91.1% previously reported by another hospital in Taiwan and the 92.8% reported by a Singapore hospital [7,16]. The seroprevalence rates in this study were lower than those reported by Japanese healthcare institutions, which ranged from 94.7% to 97.4% [17,18,19]. Data on HCWs in the wider Southeast Asian region are not available for comparison except for a small sample studied in Malaysia, which had a seroprevalence of 84.4% [20].

Routine surveillance of anti-VZV titers was implemented for the staff at this medical center in 2008, with boosters required for all HCWs with undetectable antibody titers. The seroprevalence rates after/before 2007 were marginally significantly different after the vaccination policy changed. Follow-up testing of varicella IgG titers was done 1 month after booster administration following the healthcare personnel vaccination recommendations at this hospital [15]. Between 70% to 90% adults given two doses of varicella vaccine 4–8 weeks apart are expected to develop protective antibodies, and the antibody titer can be only maintained for 7–10 years, not lifelong [1]. Whether additional vaccinations are needed later is subject to further observation [15,20].

Past hospital policy provided free vaccination of HCWs in conjunction with outbreak investigations and annual screening. The significant increase in seroprevalence recorded since 2008 coincided with a change in the pre-employment policy implemented in the previous year that required new employees to be serologically screened and vaccinated. In this study, the observation that administrative and nursing staff were less likely to be immune than doctors is in keeping with previous national vaccination policies that resulted in lower immunity in younger staff members. In this study, nurses and administrators were younger than the other HCWs. The results were similar to those of a previous study in Singapore in which administrative staff and service providers were less likely to be seropositive than older HCWs [16]. Varicella is an airborne infection that does not discriminate between vocations [4]. Consequently, both clinical and nonclinical members of the workforce are vaccinated at this hospital.

The interventions of the infection control department in hospital could help to build increased trust in the effectiveness and safety of vaccines and in the vaccine policy, ultimately enhancing the contribution of all staff to increase vaccination rates and build vaccine acceptance and resiliency in the face of the anti-vaccine lobby in HCWs and, subsequently, in the general population [21]. The hospital director must recognize the importance and impact of a successful HCW vaccine policy, distinguishing “forced vaccination, free vaccination, and convenient vaccination” and including a systematic tracking plan [15].

To prevent the transmission of varicella to healthcare workers, according to the Advisory Committee on Immunization Practices (ACIP)/Centers for Disease Control and Prevention (CDC), whom recommend that all healthcare workers (HCWs) be immune to VZV [1,22,23]. Therefore, the first step in the prevention of healthcare-associated transmission of VZV in a local screening program is to minimize the number of susceptible HCWs [12,24]. The history of vaccination or immunity for VZV should be determined at the time of initial employment in all HCWs. Serologic screening, such as VZV IgG, before vaccination is indicated for persons with no or uncertain history of varicella [21,25,26].

This study has several limitations and should be interpreted with caution. Firstly, it included only HCWs from a single hospital. The results may not be generalizable to HCWs at other hospitals in Taiwan. Nevertheless, this is the first study to evaluate varicella immunity after the implementation of national and HCW vaccination. The study population was large and comprised all HCWs regardless of job title. Ultimately, the findings will help to guide the development of a local policy to identify HCWs who are no longer immune and to develop immunization recommendations. Secondly, we did not evaluate the duration of HCWs’ occupational history, the possibility of longer occupational history provided more opportunities to reinforce their immunization status, Thirdly, Some HCWs were found to be seronegative 1 month after two doses of VZV vaccine. We did not follow up after 6 months because according to the recommendation from ACIP: Post-immunization serology is not recommended since: 1. Commercial tests may lack the sensitivity to detect the lower antibody levels associated with vaccination compared with natural infection. 2. Sensitive tests that are not generally available have indicated that 92% to 99% of adults develop antibodies after two doses of varicella vaccine [21].

## 5. Conclusions

The varicella seropositive rate of the HCWs in this hospital was not high enough to prevent outbreaks. Evidence of immunity for HCWs includes one of the following: (1) written documentation with two doses of vaccine, (2) laboratory evidence of immunity or laboratory confirmation of disease, (3) diagnosis or verification of history of varicella or herpes zoster by a healthcare provider [2]. Confirmation of the VZV IgG status of all HCWs and the vaccination of those who are susceptible is recommended for effective prevention of nosocomial varicella infections. A review of 10 years of data revealed that some elements of the workforce benefited from pre-employment screening and vaccination. Future studies should include evaluation of immunization records to determine whether immunity has been acquired naturally or by vaccination. In addition to mandatory vaccination of new employees, we want to decrease the residual population who are not immune to varicella. Those individuals are at risk of infection and pose an infection risk to our patients and are vulnerable to complications of varicella. We have achieved improvements in seropositivity but more needs to be done to ensure that staff, patients, and hospital visitors are protected from this vaccine-preventable disease.

## Figures and Tables

**Figure 1 ijerph-16-03544-f001:**
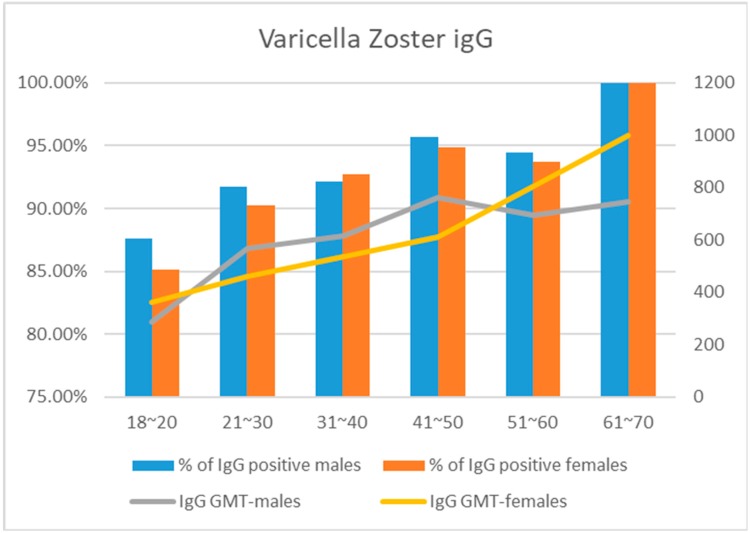
Proportion of varicella zoster immunoglobulin G (IgG)-positive subjects and varicella zoster IgG geometric mean titer (GMT), by age group and gender. Orange columns: % of varicella zoster IgG-positive females (*p* < 0.001). Blue columns: % of varicella zoster IgG-positive males (*p* = 0.310). Yellow line: Varicella zoster IgG GMT in females (*p* < 0.001). Gray line: Varicella zoster IgG GMT in males (*p* < 0.001) (Chi-squared test; linear regression).

**Table 1 ijerph-16-03544-t001:** Distribution of enrolled subjects by gender.

Age Group	Female		Male		Total	
	*n*	%	*n*	%	*n*	%
18–20	748	13.30	137	8.11	885	12.10
21–30	3632	64.57	1146	67.85	4778	65.33
31–40	787	13.99	305	18.06	1092	14.93
41–50	330	5.87	70	4.14	400	5.47
51–60	112	1.99	18	1.07	130	1.78
61–70	16	0.28	13	0.77	29	0.40
Total	5625		1689		7314	

Chi-square = 66.426; *p* < 0.0001.

**Table 2 ijerph-16-03544-t002:** Crude seroprevalence by vocation (*n* = 7314).

Vocation	Age (Years) (Mean ± SD)	*p*-Value	*n*	Seroprevalence in % (Binominal 95% CI)	*p*-Value
Nurses	24.82 ± 0.42	<0.001	2826	88.9 (87.7–90.0)	<0.001
Doctors	27.74 ± 0.38		1394	92.8 (91.4–94.2)	
Examination	27.97 ± 0.12		481	93.2 (91.0–95.4)	
Preventive and long-term care service	33.06 ± 0.24		226	91.5 (87.9–95.1)	
Administration	25.87 ± 0.13		1702	91.0 (89.6–92.4)	

**Table 3 ijerph-16-03544-t003:** Multiple logistic regression of immune status after national policy (*n* = 7314).

Variable	Odds Ratio	*p*-Value	95 % CI	
			Lower	Upper
Gender				
Female	1.000	-	-	-
Male	0.968	0.775	0.774	1.210
Birth-group				
After 1 January 2003	1.000	-	-	-
Before 31 December 2002	2.506	<0.001	1.902	3.515
Age-group				
18–20	1.000			
21–30	1.703	<0.001	1.349	2.151
31–40	2.406	<0.001	1.683	3.438
41–50	5.551	<0.001	2.671	11.536
51–60	5.734	<0.001	2.261	14.542
61–70				
Vocation				
Doctors	1.000	-	-	-
Nurses (ER, clinics)	0.579	0.001	0.414	0.811
Nurses (wards)	0.661	0.002	0.507	0.861
Examination	0.975	0.902	0.650	1.462
Preventive and long-term care service	0.747	0.259	0.450	1.240
Administration	0.690	0.006	0.531	0.898

**Table 4 ijerph-16-03544-t004:** Univariate logistic regression of immune status after hospital policy (*n* = 7314).

Variable	Odds Ratio	*p*-Value	95 % CI	
			Lower	Upper
Year				
(a) 2003–2007 (before policy)	1.000			
2008–2018 (after policy)	1.456	0.052	0.967	2.345
(b) 2003–2007 (before policy)	1.000	-	-	-
2008	2.036	0.046	1.283	4.216
2009	1.538	0.233	0.758	3.121
2010	1.392	0.361	0.685	2.829
2011	1.545	0.226	0.764	3.126
2012	1.577	0.207	0.777	3.199
2013	1.852	0.098	0.893	3.844
2014	2.088	0.049	1.100	4.360
2015	1.774	0.119	0.863	3.645
2016	1.477	0.288	0.720	3.028
2017	1.234	0.564	0.604	2.519
2018	1.056	0.889	0.491	2.271

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
