# Peer review of "Varicella Seroprevalence in Healthcare Workers at a Medical Center Following Changes in National and Local Hospital Vaccination Policies"

_ijerph, 2019, doi:10.3390/ijerph16193544_

Round 1

Reviewer 1 Report

Thank you for the opportunity to review this manuscript. Overall, this is an interesting topic where the objective was to assess the varicella seroprevalence among HCWs after the implementation of national and hospital HCW vaccination policies in Taiwan.

I don't feel qualified to judge about the English language and style. There is good data in this study, however, the presentation and interpretation of the method and discussion section need significant additional thought. Here are some suggestions to improve the manuscript:

*In the Abstract section, please consider highlight the purpose of the study.

The introduction provide a small background, please include some new relevant references (last years). You cite some articles but it would be helpful to the reader to have a more specific overview of the findings. 
Page1, line 41: I suggest you change "11,000 cases" to "11000 cases"

Objective. You have said: “This study assessed the varicella seroprevalence among HCWs after the implementation of national and hospital HCW vaccination  policies in Taiwan to help guide the development of a local screening program”. Have you discussed about “to help guide the development of a local screening program”? Maybe, you should give some dates or recommendations about the development of a local screening program..in the Discussion section.

Methods. Please clarify the research design; the eligibility criteria (inclusion and exclusion criteria). The procedure and if patients provided written informed consent?. Please, consider to expand your description of sample accordingly the power analysis used to determine the number of potential individuals that you needed to include in your study. This is not sufficiently addressed in this manuscript. The management of missing data is lacking.

The results. Please, report the same numbers of decimals in the datas (ei, average, SD…)

Discussion. Start the discussion by reporting your own findings from the present study and then, after that, you put it in perspective of other available and recent research. Do not repeat your own results (numbers) in the discussion if not absolutely necessary to make a certain point. Use descriptive language instead. You cite some articles but it would be helpful to the reader to have a more specific overview of the findings, you have only 18 references in all the manuscript.
Please include a clinical implications section: to help guide the development of a local screening program

 Good Luck!!

Reviewer 2 Report

Sir, first at all thank you for the opportunity to review your interesting paper about "Varicella seroprevalence in healthcare workers at a medical center following changes in national and local hospital vaccination policies". 

Even though the main topic of this research paper (i.e. seroprevalence rates in HCW from a specific medical center) may appear of limited and somewhat reduced interest for the international readers, the importance of VZV as an occupational pathogen stresses the relevance of all reports addressing preventive measures, including vaccination policies specifically tailored to  the occupational settings.

Still, the present paper should be extensively improved in order to obtain a full publication on IJERPH.

First at all, a methodological concern about statistics and subsequent results. Authors have reported that immune status significantly improved across the age groups (see Figure 1, Table 2 and Table 3): as the variables they assessed included age, occupational status, being born after/before the changement of policies, a potential colinearity may impair the soundness of their results. I'm certain that the Authors have accurately and preventively modeled their regressions analysis models in order to avoid the effects of colinearity on their results, but I warmly recommend to improve methods section in order to more appropriately explain the measures they applied. Similarly, I do suggest the Authors to collapse analyses reported in table 4 as "workers enrolled before / after 2008" in order to more appropriately show the effects of local policies for occupational vaccinations.

Second, Authors should be aware that their results suggest that workers from younger age groups have more an inappropriate immunization status towards VZV than older ones. This is reasonable, as older people have had more opportunities to be in contact with the wild-type virus, eventually reinforcing their immune status, than younger workers - particularly after the policy shift of 2004, but it may appear inconsistent with the reported changes of occupational policies reported after 2008.

In other words, the effect of national policies on immunization rates of younger workers is clearly indirect, and its interplay with local policies should be more extensively formulated both in the introduction and in the discussion section - otherwise, results would appear somewhat contradictory and inconsistent. 

Third, please be aware that the categorization in the reported age group is unclear, and such flawless affects the soundness of your results. More specifically: you included following age groups, 11-20, 21-30, 41-50, 51-60, 61-70. This is appropriate for an ecological study on the general population. But your study is focused on Occupational Settings, and it is unclear as an occupational study on HCW may include workers aged 11-20 years. I think that the reported group did actually include some mainly nurses and some administrative workers at their enrollment. However, Authors should be aware that it would both inflate colinearity (as such professionals are expected to be oversampled among younger age groups, while MD are similarly expected to be oversampled among older age groups) and include another variable, that is the potential previous occupational exposure of the workers, as you actually show data from VZV naïve HCWs (whose opportunities to interact with the VZV have been impaired by the National Policies) and HCWs having a very long occupational history (i.e. more opportunities to reinforce their immunization status). Otherwise (i.e. the reported groups rather include non-workers or controls) please reformulate methods section accordingly.

Fourth, I warmly suggest to reformulate the closing statement (“more needs to be done to ensure that staff, patients, and hospital visitors are protected from this vaccine-preventable disease”) including specific suggestion / recommendations based on the reported results. Discussion and conclusions may be improved by assessing other international experiences from occupational vaccination services and experiences (for example, but absolutely non limited to: https://www.ncbi.nlm.nih.gov/pubmed/28661516; https://www.ncbi.nlm.nih.gov/pubmed/30181734 ; https://www.ncbi.nlm.nih.gov/pubmed/15635972 ).

Eventually, even though the English text is largely appropriate, I would suggest to revise the introduction and the discussion in order to make both more straightforward. For instance, in the introduction section Authors wrote:

“It is important that the immunity of the hospital staff to varicella zoster virus (VZV) be monitored in order to decrease the risk of nosocomial infection and infection to the staff and visitors”, then give some appropriate snapshots about VZV in the Healthcare settings, describing general national policies and eventually describing occupational policies. I think it would be more simple starting with a general description of VZV burden of disease, then focusing on occupational settings, eventually explaining why it is so important monitoring hospital staff about VZV. Please be aware that the latter is only a personal suggestion, that Authors should not feel forced to apply.

Round 2

Reviewer 1 Report

Thanks for your careful revisions of the manuscript. The manuscript is by now much clearer

Reviewer 2 Report

Sir,

thank you again for the opportunity to contribute to IJERPH.

Authors have extensively implemented all corrections we suggested, and similarly addressed all our doubts about study design and data presentation.

In my opinion, the present version of the paper deserves a full publication.

I've only a very small suggestion for table 3: because of the design of the table, readers may misunderstand that rows for age groups 51-60 and 61-70 in fact were collapsed in a single 51-70 years age group. As a consequence, I suggest either to collapse in a single label (i.e. 51-70) or in a single row (i.e. "51-60 + 61 - 70").

I've no further recommendations.